# Preprint articles as a tool for teaching data analysis and scientific communication

**Lisa Z. Scheifele** [1]*, **Nikolaos Tsotakos** [2], **Michael J. Wolyniak** [3]

**1** Department of Biology, Loyola University Maryland, Baltimore, Maryland, United States of America,
**2** School of Science, Engineering, and Technology, Penn State Harrisburg, Middletown, Pennsylvania,
United States of America, **3** Department of Biology, Hampden-Sydney College, Hampden-Sydney, Virginia,
United States of America

* lzscheifele@loyola.edu

**Data Availability Statement:** All relevant data are within the manuscript and its Supporting information files.

## Abstract

The skill of analyzing and interpreting research data is central to the scientific process, yet it is one of the hardest skills for students to master. While instructors can coach students through the analysis of data that they have either generated themselves or obtained from published articles, the burgeoning availability of preprint articles provides a new potential pedagogical tool. We developed a new method in which students use a cognitive apprenticeship model to uncover how experts analyzed a paper and compare the professional's cognitive approach to their own. Specifically, students first critique research data themselves and then identify changes between the preprint and final versions of the paper that were likely the results of peer review. From this activity, students reported diverse insights into the processes of data presentation, peer review, and scientific publishing. Analysis of preprint articles is therefore a valuable new tool to strengthen students' information literacy and understanding of the process of science.

## Introduction

The advent of online publishing has ushered in new and diverse ways of sharing academic knowledge. New venues have emerged such as online open-access journals, academically-oriented Twitter feeds [1,2], podcasts [3,4], and the growing popularity of preprint servers. Collectively, these new methods seek to disseminate scientific information more quickly and more directly to scientific researchers, medical practitioners, the lay media, and the general public. They have also had a dramatic impact on the traditional publishing and peer review process; examples of changes within the traditional publishing system include the use of open review in which reviewers names and/or comments are included online with the published article [5,6], post-publication peer review [7,8], and a rethinking of the objective and subjective criteria that ought to form the basis for peer review [9]. As educators, we must ensure that our students are aware of the modern means by which scientific information is disseminated as well as the strengths and limitations of each mechanism.

Preprint servers are an important part of this new communication landscape. Preprints are complete manuscripts that have not gone through peer review but may be published to an

**Funding:** The author(s) received no specific funding for this work.

**Competing interests:** The authors have declared that no competing interests exist.

online preprint server in a parallel and independent process to journal submission. The first preprint server to be founded in 1991 was arXiv (https://arxiv.org/) which serves the physics, astronomy, computer science, engineering, and mathematics communities, followed by bioRxiv (https://www.biorxiv.org/) for the life sciences in 2013 [10] and medRxiv (https://www.medrxiv.org/) in 2019 for the health sciences [11]. Although the frequency varies by discipline, the use of preprint servers to disseminate research has increased dramatically from 2014, when there were 4,012 authors of preprint articles on the bioRxiv server, to 2018, when there were 106,231 [12]. Indeed, publication on a preprint server is increasingly becoming the first step of publishing a research article, and most manuscripts that are submitted as preprints eventually go on to be peer-reviewed and published in academic journals [13]. This broad acceptance of preprint publication has been further and dramatically accelerated by the COVID-19 pandemic as fully 2/3 of all preprints posted to medRxiv in 2020 were related to SARS-CoV2 and COVID-19 [14].

This unique publishing environment offers new opportunities for science education. Strong pedagogical approaches such as the CREATE model [15], article annotation [16], and rhetorical analysis [17] have been developed to teach students the scientific research process and how research data is communicated in journal articles. While peer review is recognized for its importance in keeping low-quality research out of the public discussion and in quality assurance [18], there are fewer methods that teach students the process of scientific publication and peer review [19,20]. This is a missed opportunity to develop the students' abilities to communicate science and to use quantitative reasoning, as stated in *Vision and Change* [21]. Indeed, initial forays in this area have found that having students engage in evaluation of their peers' work increases students' understanding of the scientific publishing process and the role of peer review within that process [22], identifies and corrects student misconceptions [23], and enhances their writing and critical thinking skills [19]. At the professional level, both making peer reviews open and accessible post-publication and mentoring trainees in the review process have been noted for their pedagogical value [24,25].

Methods that engage students in peer review as a pedagogical tool often have students act as reviewers of either published journal articles or their classmates' work. One challenge in this approach is that students struggle with interpreting research data [26], an exceptionally difficult skill, and are known to read articles passively and superficially, focusing more on the narrative of the text than on the data presented in the figures and tables [27,28]. We therefore developed a cognitive apprenticeship approach [29–31] to develop students' skills in critiquing research papers and experimental data. In the cognitive apprenticeship model, students develop skills by engaging in realistic, real-world tasks. Unlike a traditional apprenticeship, cognitive apprenticeship helps students to first reveal and then to master the cognitive and metacognitive skills and processes used by experts [31]. Central to this process are the principles of modeling, in which students see how experts approach problems, coaching, in which students are guided in reflection on how their novice process differs from the expert method, and scaffolding, in which students are provided with cognitive tasks of increasing complexity that can help them to achieve "successive approximation of mature practice" [30].

We applied the cognitive apprenticeship approach to the teaching of data interpretation and critiques by having students review articles themselves and then see how professionals critique the same paper, thereby revealing and making explicit the expert's thinking and analysis of the manuscript. This professional peer review can be deduced from changes that have occurred between the preprint and final versions of an article; therefore, by uncovering this professional assessment, students can reflect on the differences between their assessment and the peer reviewer's. Studying these professional reviews can help students to pinpoint the most important aspects of paper, to critically evaluate the quality of the data, and to identify areas

where information is not effectively communicated. Although there are other tools that can help students to see how professionals critique papers (for example peer reviews that are posted online), having students compare the preprint and final articles themselves gives them an *active* and engaging way to discover weak areas in the original paper and to reveal this information for themselves. Importantly, this method can be employed for research papers regardless of their topical content, and we found success with this method using 10 different articles at three different institutions.

We hypothesized that the detailed comparison of the preprint and published, peer-reviewed versions of an article would help students to develop skills in the analysis of research articles as well as provide students with unique insights into the research process. Indeed, we found that comparison of preprint and final versions of a paper: (1) helped students to critique research data by uncovering weaknesses in the preprint version that they initially failed to identify, (2) helped students to appreciate the role of peer review in ensuring publication quality, (3) helped students to understand and critique the process by which scientific results are communicated.

## Methods

We determined the educational benefits of analyzing preprint articles for students in three courses during the Fall 2020 semester: BL481: Biology Research I at a mid-sized private comprehensive university, BIOL 451: Biology of RNA at a mid-sized public comprehensive university, and BIOL314: Medical Genetics at a small private liberal arts college for men as well as one course during the Spring 2021 semester: BL322: Synthetic Biology with Lab at the same mid-sized private comprehensive university. BL481 is an independent 3-credit research course in which students worked closely with a faculty research mentor (3 students enrolled and participated in this study). BIOL451: Biology of RNA (7 students enrolled and participated in this study), BIOL314: Medical Genetics (13 students enrolled and participated in this study), and BL322: Synthetic Biology with Lab (24 students enrolled, of which 6 chose the option to participate in this study) are upper-level courses for students in the biological sciences Two students did not complete all three parts of the project and their work was not included; in total 27 students from the 3 institutions participated in this study. They ranged from sophomore to seniors, with 2 being sophomores, 8 being juniors, and 17 being seniors. Because of the COVID-19 pandemic, courses at the private comprehensive university were being taught entirely online in Fall 2020, while classes in the spring semester and at the other institutions were held in-person but socially distanced.

A strength of the learning process described here is that it is independent of the content of the research being discussed. At each institution, different research preprints and final articles were therefore used. At the private comprehensive university, students each analyzed a different pairing of preprint and published paper; articles focused on genome rearrangement in yeast [32,33], reconstruction of the SARS-CoV2 genome [34,35], synthesis and mutation of a bacteriophage genome [36,37], gene expression burden [38,39], design of RNA regulators [40,41], noninvasive prenatal tests [42,43], programmable gene regulation [44,45], and metabolic engineering [46,47]. At the public comprehensive university, all students analyzed a paper on the topic of long noncoding RNA [48,49], and at the liberal arts college, all students analyzed a paper on molecular analysis of neurosensory syndrome [50,51].

In designing our study methodology, we followed three steps of the cognitive apprenticeship: mimicking, modeling, and coaching, For each of these stages, students completed and submitted worksheets as separate assignments: (1) for mimicking professional practice, students analyzed the preprint version of the article (S1 File), (2) for modeling, students compared the preprint and final published version of the article (S2 File), and (3) for coaching,

students reflected on the learning experience (S3 File). Each stage proceeded sequentially, and students were not given the next assignment until the previous one had been submitted. Students met with the instructor and discussed each part of the assignment close to the day on which it was submitted. Instructors prompted students to discuss their findings, what aspects of the assignment were easy or difficult for them to accomplish, and what was unexpected or surprising in their analysis of the preprint or final paper. All parts of this study and assessment methods were approved by the Institutional Review Board of Loyola University Maryland (HS-2021-013) and written consent was obtained from all participants.

To analyze whether analysis of preprints and final papers enabled students to better critique research data, we first analyzed student worksheet part 1. For each instance in which students made a critique of the preprint article, that critique was classified by a faculty grader as falling into one of the following 4 categories: (1) additional repetition, controls, or an extension of the experiment is needed, (2) statistical test(s) are needed, (3) clarification of either the text or depictions of the data is needed, (4) inappropriate conclusions have been drawn from data. These data were compiled and both the average number of critiques per student and the range were calculated. Student comments regarding formatting (font size, figure formatting, or how graphs are labeled, as examples) were not included as critiques, but comments indicating that the data was difficult to interpret or needed additional labeling were included as critiques. For student worksheet part 2, we first began by noting any changes that students identified between the preprint and final versions of the results section of the paper; these changes were sorted into the same 4 categories described above so that the student critiques of the results from part 1 could be compared to the actual changes in the final paper that were identified in part 2. For student worksheet part 2, we analyzed not only the results section but also the introduction and discussion sections. Changes in these sections were categorized as to whether they were additions, deletions, or changes to the information and whether they occurred in the background information, experiments and their results, or conclusions. To analyze the student reflections in part 3, we used coding analysis for qualitative data according to the methods described in Saldana [52]. First level coding was performed using three different methods: descriptive, in vivo, and values coding, and axial coding was used for second-level coding and inductive analysis of themes.

## Results

### Role of the preprint and final paper analysis in helping students to critique research data

Students in our classes had a range of academic preparation and some had little previous exposure to primary literature. We therefore wanted students to first become familiar with the research paper that they were analyzing. Students' first worksheet therefore began with a self-guided analysis of the introduction section of the preprint paper in which they were asked to identify the main problem being addressed, the importance of answering that question, the most important background information, and the hypothesis being tested (A copy of the student worksheet is available as S1 File). In this first part of the assignment, students also analyzed the discussion section to identify the major conclusions, restate the importance of the work, and reflect on any future directions indicated by the author(s). Although we did not analyze these portions of the worksheet for this study, they provided students with the deep familiarity that they required to analyze the research results.

We then determined what critiques students made of the preprint (S1 File; an example of a completed student worksheet is included as S4 File). In this exercise, students are therefore acting in the role of peer reviewers for the preprint; because most of the preprints were posted on

the preprint server within days or weeks of when they were submitted to a research journal, the preprint should be nearly equivalent to a journal article at the time of submission. On average, students provided 2.04 critiques of the results section of the preprint article that they analyzed (Table 1); four students had no critiques while all others identified between 1 and 9. Students' most common critiques included the need for additional clarification of the data or its interpretation (average of 0.96 per student) and the need for additional repetition, controls, or extension of experiments (average of 0.88 critiques per student). Students were less likely to indicate that statistical tests were used inappropriately or insufficiently (average of 0.08) or that data was interpreted incorrectly (average 0.13). While the number and types of weaknesses present will certainly vary depending on the preprint article selected, students were most likely to identify places where they feel clarification is needed; while this may be unsurprising for novice readers, it provides a preliminary suggestion that students may not be equally adept at identifying all types of manuscript weaknesses.

We next determined whether students were able to compare the preprint and final versions of the article to identify changes (S2 File; an example of a completed student worksheet is included as S5 File); these changes presumably represent weaknesses in the preprint that were identified by professional scientists (Table 1). In this analysis, we did not include any superficial changes to formatting or figure quality; we also did not include changes to factual information or data values as these might be the result of error correction by the authors that a peer reviewer would be unlikely to anticipate. Central to our analysis is the assumption that most of the remaining substantive changes between the preprint and final versions of the article are due to the peer review process; while some might be due to changes initiated by the authors either independently of the peer review process or due to community feedback following the publication of the preprint, these changes still represent ones initiated by science professionals to remediate weaknesses in the preprint, and we therefore believe that these are worth including in our tabulations. Students identified an average of 3.38 changes between the results sections of the preprint and final versions of the paper (range of 0–10 changes). The most common changes were to the conclusions drawn from the data (average of 1.04 changes), clarification of how the data is presented or described (average of 0.67 changes), and the need for additional experiments or changes to how the data is presented (average of 1.58 changes). There were therefore a greater number of changes made between the preprint and final

**Table 1. Suggested and actual changes between preprint and final versions.**

|  | Student Critique of Preprint (Part 1) | | Student Analysis of Final Paper (Part 2) | |
|---|---|---|---|---|
|  | Avg. | Range | Avg. | Range |
| Additional repetition, controls, or extension of experiment | 0.88 | 0–5 | 1.58 | 0–5 |
| Statistical test needed | 0.08 | 0–1 | 0.09 | 0–1 |
| Clarification needed | 0.96 | 0–3 | 0.67 | 0–4 |
| Inappropriate conclusions drawn from data | 0.13 | 0–1 | 1.04 | 0–3 |
| Total number of critiques | 2.04 | 0–9 | 3.38 | 0–10 |
| Changes in common between parts 1 and 2 | 0.08 | 0–1 |  |  |
| Changes in part 2 not identified in part 1 | 3.25 | 0–10 |  |  |

The average and range for the number of critiques of the preprint version of the article (based on the worksheet Student Critique of Preprint Part 1; S1 File) and the average and range for the actual number of changes that students identified between the preprint and final versions of the paper (based on the worksheet Student Analysis of Final Paper Part 2; S2 File).

versions (average of 3.38) compared to the number of critiques (ie. suggested changes) made by students (average of 2.04).

For some types of changes, the number of changes that occurred was close to the number suggested by students (for example, students identified an average of 0.96 instances where clarification was needed while an average of 0.67 clarification changes were actually detected in the final version). But importantly, the identity of these changes differed. Indeed, on average there were only 0.08 changes per paper that were common between the student critique and the actual changes that were made between the preprint and final versions, and only 2 students critiqued the preprint in a way consistent with changes actually made to the paper. This indicates a significant difference in the critiques made by the students (in part 1 of their assignment) and by science professionals (changes between the preprint and final versions that students identified in part 2), as could be expected [28]. In all cases except one, students were able to identify changes (an average of 3.25 per paper; range of 0 to 10) between the preprint and final versions that did not correspond to an issue that they identified in their own critiques. This exercise was therefore able to help students uncover for themselves areas of weakness in the original preprint that they had missed; it is therefore another valuable pedagogical tool to help upper-level students learn to rigorously analyze research data.

While our primary objective was to use this technique to help students learn to analyze research data and data presentation, we used the opportunity to have students also analyze changes to the introduction and discussion sections of the paper as this could be an additional tool to help them to understand the scientific publishing process. Because students are not experts in the discipline, we did not think it reasonable to ask them to perform their own critiques of the introduction and discussion sessions of the preprint, so we only asked them to identify changes between the preprint and final versions (Student worksheet part 2, S2 File). Students identified additions, deletions, and changes of information, with additions of information (perhaps unsurprisingly for the peer review process) being the most common (Fig 1). In each instance, students were able to identify changes between the two versions, with an average of 7.58 changes (range of 2–20) identified per article. This activity therefore provided students with additional insight into the types of changes that can be introduced and requested by reviewers through the peer review process.

## Role of the preprint and final paper analysis in helping students to appreciate the role of peer review

Because our method was effective in having students use an active learning approach to identify evidence of peer review in the final versions of published papers, we wanted to extend this analysis by determining whether this process also helped them to learn more about the role and importance of peer review. To determine this, the third part of the student assignment asked them to respond to several questions including: (1) What surprised you most when you compared the preprint and final versions of the paper? (2) Is it easy or hard to distinguish which additions were made because the authors had more time to acquire more data and additions that were made due to requests from the peer reviewers? How do you think you can you tell the two apart? (3) What, if anything, did you learn about presenting and interpreting research data from this activity? (4) Do you think that this paper and the publishing process that it went through achieved an appropriate balance between getting research results out quickly versus the need for high-quality, complete, and reproducible research?

When comparing the preprint and final versions of the paper, some students were surprised at the number of changes between the two versions (9 students) while others commented on how few changes there were between the two (13 students). Interestingly, sometimes these

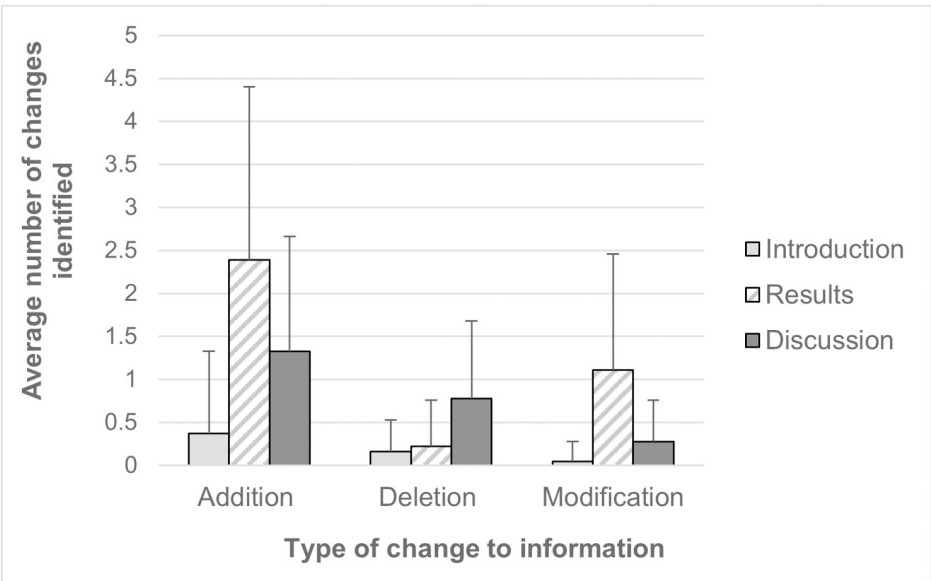

**Fig 1. Average number of changes between the preprint and final versions.** The number of changes as identified by students in the introduction (light gray bars), results (hatched bars), and discussion (dark gray bars) sections. Error bars represent the standard deviations for each value.

comments came from the same student! In these cases, students were surprised that there were few changes, but that the changes were as substantial as the addition of an entire experiment. For example, a student commented:

> This would have taken considerably more lab work so would have involved a significant effort by the research team after putting out the preprint. I was even more surprised as this experiment was probably not absolutely vital to validate the research performed. However, I do believe it probably does improve the overall strength of the research.

> *-male student, senior class*

Students were also surprised at the types of changes that they observed; in addition to adding new experiments, these included changes that clarified information or emphasized new topics, large-scale rearrangements of entire sections of the paper to restructure the argument, the fact that the majority of changes were additions of information, and changes to how figure were presented and formatted that led to better emphasis on specific aspects of the data and better integration of data figures with interpretation of results.

In terms of whether these changes were the result of the peer review process or whether they were due to the authors having more time, the students were evenly divided as to how easy that was to determine. Changes that they felt were indicative of reviewer comments included those that provided greater clarity, better flow to the paper, additional background information, removal of uncorroborated claims or speculation, and changes to the interpretation of the data to include more uncertainty. Additionally, students predicted that reviewer input could be detected when experiments in the preprint had noticeable defects that were fixed in the final version, when statistical tests were added, and when experiments were added that provided more consistency.

Personally, I learned that the scientific publishing process can cause highly noticeable changes to one's paper in the results, interpretation of results, and discussion portions of the paper. . .. In particular, this paper had shown me the difference in quality in the explanation of the results and in discussion of the overall paper. Furthermore, I learned that the requests of peer reviewers can often work to help a paper be more clarified in its explanations. Many times throughout the paper, I saw an improvement in the wording of certain interpretations that allowed me to better understand what they're [trying] experimenting with and the significance of the results.

*-male student, sophomore class*

On the other hand, they predicted that changes to data values or the addition of additional crucial data was the result of the researchers having more time. Responses to this question therefore indicate that students understand the role of peer reviewers in ensuring that data is interpreted properly, that the paper is clear to read, and that defects in experimental procedures or data presentation are rectified.

Student reflections therefore demonstrate that this exercise helped them to better understand the process of peer review, but there were other significant components of presenting and interpreting research data that they also learned (Table 2). They mentioned that the exercise helped them to better understand how to read scientific papers (including the need to pay attention to detail and the utility of jumping around between multiple sections of the paper), to understand the importance of precise wording in scientific writing, to properly use and present scientific data, to interpret data, and to understand how the process of scientific publishing and peer review occurs. Students therefore reflected on incredibly diverse aspects of the scientific process. While this exercise intended to help them to build skills in data analysis and to understand the role of peer review, they clearly gained a greater breadth of understanding about data presentation, interpretation, and publishing.

Student responses regarding what they learned about the process of peer review, writing journal articles, presenting data, and interpreting research data are presented (second column) along with categorization of those responses (first column). Student responses are summarized except where direct quotes are indicated.

## Role of the preprint and final paper analysis in helping students to understand scientific communication

To assess the effect of this exercise on students' understanding of scientific communication, students were asked to respond to several questions in part 3 of the assignment: (1) Do you feel that the level of attention that this non-peer reviewed article received was appropriate? (2) Do you think that preprint servers are a good way to get results out in a hurry? Should we maintain this process, or are there changes that should be implemented? (3) What, if anything, did you learn about the scientific publishing process from this activity?

In terms of whether the level of attention that the paper received was appropriate, whether the process should be maintained, and if any changes should be implemented, students pointed out several reasons why their research articles should have received a good level of attention (Fig 2). They felt that preprint servers are a good way to get "raw, unfiltered data out in a hurry". This was especially true for research with strong public health implications:

I think preprints are a good idea to some extent. Waiting for all the data to get published would take way too long, especially for topics that are greatly affecting the world like SARS-CoV-2. Having preprints allows for the information to be shared quicker and can

**Table 2. Results of student reflections on the preprint analysis activity.**

| | |
|---|---|
| Reading papers | The changes focus the reader on important points. |
| | "There are many bits of information that I miss while reading." |
| | The figures and text complement each other. |
| | Each figure aimed to prove one claim. |
| | It reinforced the importance of figures relative to the text interpretations. |
| | The need to scrutinize data |
| | Reading scientific literature is a skill that must be developed over time. |
| | Can skip around the paper when reading |
| | Scrutiny is required to fully grasp the meaning of figures and data. |
| | "I also learned that not fully understanding certain vocabulary in a paper does not mean I can't understand the bigger picture." |
| Language in scientific writing | The importance of wording |
| | Every detail has to be explained |
| | Write to the reader; you need to communicate effectively. |
| | It's best to be simple and concise; leave interpretation to the discussion. |
| | Need to communicate findings effectively and to be as specific as possible to prevent misinterpretations |
| Use and presentation of data | Experiments build on previous work. |
| | Scientists use multiple pieces of evidence to support claims. |
| | "I found that there is a lot that goes into making a claim, in effort to come up with data that is consistent and considers all aspects of the claim being made." |
| | "Need to. . . .obtain not only enough, but quality experimental results" |
| | Scientists present both the strongest research and also weaker data that provides context. |
| | Scientists use multiple methods to prove one claim. |
| | Need to present data effectively |
| | The way that the data is presented is important. |
| | Data may not fit the scientists' theory. |
| Interpretation of data | "Sometimes you don't interpret the data the same way the scientist who did the experiment." |
| | "Even if the overwhelming preponderance of evidence is in your favor, you must provide alternatives that could explain the findings of the data." |
| | Initial thinking about the data may change and may become more advanced as the publication process advances. |
| | Judgement is required: even researchers don't always know what is important to support their thesis. |
| Scientific publishing and peer review | How specialized certain scientists, and their fields of interest are |
| | In final paper, "the information is clearer and more focused". |
| | Data is shared long before the publication date. |
| | Going from the preprint to the final is a "shaving down process". |
| | "I learned just how important details can be when publishing scientific research. Even the smallest things were commented on by the reviewers and would prevent the piece from being published until corrected." |

help other researchers expand their knowledge of the topic to propel their research forward.

*-female student, junior class*

Students expressed that preprints brought attention to the research results, and that this could be beneficial both for papers in highly popular and impactful fields, like SARS-coV2 and

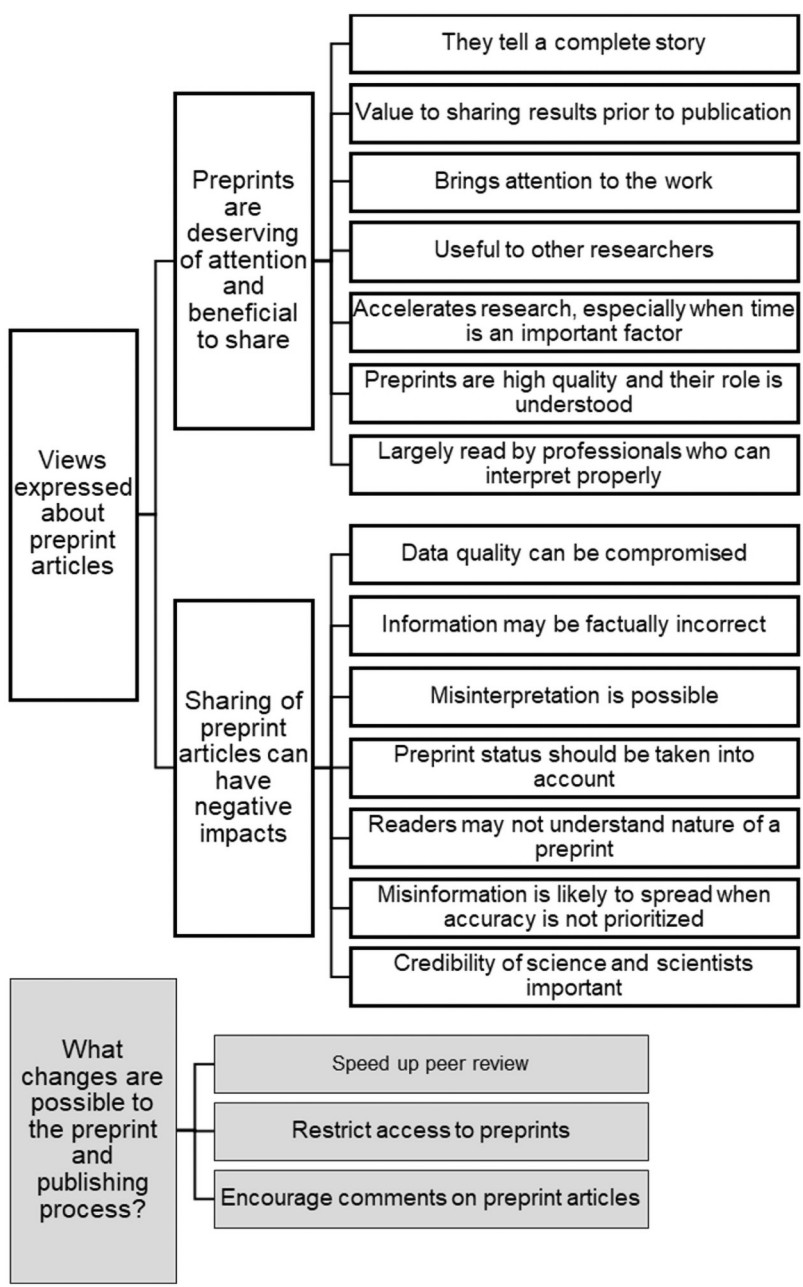

**Fig 2. Student views of preprint articles and what changes could be implemented in the preprint process.** Student responses to the worksheet questions of whether "preprint servers are a good way to get results out in a hurry", whether "the [preprint] publishing process . . . achieved an appropriate balance between getting research results out quickly . . . versus the need for high-quality, complete, and reproducible research?" and whether "we [should] maintain this process, or are there changes that should be implemented?" Representative codes resulting from qualitative analysis are depicted.

COVID-19 currently, as well as for papers in more niche fields that would otherwise receive little notice. The many benefits to getting results out quickly was noted, including the ability of scientists to "mark their territory", show how far their research has progressed, prevent other scientists from duplicating their research and wasting time, and providing a venue for

scientists to gain pre-publication feedback (a "less condemning preview for authors to receive criticisms"). This was seen to be important for propelling research forward, and students recognized that the results could be useful for other scientists, even if the data were not perfect. These imperfections were noted to be tempered by the fact that the work is mostly viewed by professionals; because the majority of papers don't grab the attention of general public, students expressed that scientists understand the stage of development of preprint papers and can interpret the results in light of the fact that they are not yet peer-reviewed. The fact that there were few differences between the preprint and final versions of the article reassured students that the preprint versions were of high quality. Some also noted that it was the responsibility of the authors to submit high-quality work ("ability to produce good data is dependent on the researchers") and to understand the nature of preprints when using the data ("Most people could reason and understand that a paper still under review isn't the best evidence to act upon"). Students also expressed clear, although differing, criteria for whether preprints should get a high level of coverage; these criteria included the quality of the work, if it was peer-reviewed, and if it had public health relevance.

Students were asked to reflect on whether their preprint article received an appropriate level of attention by examining the metrics on bioRxiv, which consist of the number of times the HTML and PDF versions were downloaded and how many times the preprint was tweeted or read on Mendeley. Although they differed in their degree of agreement, most students (20 of24) felt that the level of attention generated by the preprint articles was appropriate (Fig 2), with 3 of these feeling that their preprint should have received more attention. In contrast, 4 students felt either moderately or quite strongly that the preprints received too much attention. These students were all concerned about potential errors in the research, with one student saying about the article they reviewed that it was "lucky the research was well done". Five students indicated that preprints were more likely to have incorrect information compared to peer-reviewed articles. This could "bring[s] a lot the false information to the real world" and "may be a source for providing misinformation". The 4 students with the strongest objections, as well as some of the students who still felt the level of attention was appropriate, were worried about future research being planned based on research results disseminated through preprint articles. Numerous students suggested some modifications that could improve the process (Fig 2). One suggestion was that preprints should not be cited by other research articles and that this standard could curtail new research being based on potentially faulty preprint data.

Students also expressed concerns about the nature of preprint articles being properly understood. Responses indicated that while students understood the value to sharing results prior to publication, they also recognized that one cannot assume that all data is of equal quality (i.e. that all data is "correct"). Some expressed concerns that the public might not understand the review process that the paper has or has not been through and suggested safeguards such as the addition of a disclaimer that the article is still under review so that the information could be "taken with a grain of salt" or a delay before publishing the preprint version on the online server. Several made distinctions between different types of reader and expressed less concern if the preprint was read by scientists than if it was read by the general public or media. In part 3 of the student worksheets, we asked students to note the preprint's metrics including how many times it was downloaded or tweeted; a few students referred to the gatekeeping function of media and highlighted that the tweets referencing the preprint came from media accounts. There was concern that "the media often doesn't understand the process" and one student even worried that the media doesn't want to "note the caveats about the data which often steals the thunder they are trying to harness". The most sophisticated students therefore noted the role of the media as an interpreter of scientific information, although one with its own priorities and biases.

**Fig 3. Student learning gains resulting from the preprint analysis activity.** Student responses about "what, if anything, [they learned] about presenting and interpreting research data" and "about the scientific publishing process" were analyzed and representative first-level (bullet points) and second-level (title) codes are shown.

Finally, we asked students to reflect on what, if anything, they learned about the scientific publishing process from this activity (Fig 3). Students learned about the mechanics of the scientific publishing process that they were previously unaware of, including how long the processes of conducting research and publishing take. One commented that they learned:

> how much detail goes into publishing papers. In science classes. . .our extent of learning goes into one lab class where we finish our experiment. Researchers spend months and years trying to gather information and create new discoveries to benefit their respective communities.
>
> *-female student, senior class*

It was noted that the revision process involves both large changes, such as the addition of new experiments and ideas, as well as much smaller edits. In fact, "some of the most significant changes do not need to be huge", and "these changes are not apparent at first glance but can have a large impact on the paper." Most importantly, students seemed to have learned the role and importance of peer review. They noted that the process improved the quality of the finished product. "It took a whole year to perfect this paper which really shows they care about perfecting their research." One student also commented on the high standards required for publication in a scientific journal: "I learned that the standard to publish in recognized journals is very high and this paper serves as a great example of how minor changes allow authors and researchers to meet this standard." Students also appreciated the function of peer review

in ensuring the clarity of the paper: "Publishing really makes an author focus on how the paper is received by the reader". Impressively, one student made the incisive observation that "you will always have to interpret your data", and others noted the role of peer review in ensuring that data is appropriately explained and interpreted.

Students' insight into the publishing process also included their observations about the many individuals and interests that have a significant influence on the publishing process. They noted the use of multiple reviewers and the need to "cater to the requests of reviewers" through a process that was "never a straight path" and that involves "back and forth during the review process". The competitive nature of the process was also commented upon: "What was most surprising to learn was the amount of influence that publishers and editors have over an article's ability to be published", as well as the political nature of publishing and the need to "establish[ing] territory". One of the articles analyzed involved human subjects; because the final version had fewer patient details than the preprint, one reader commented upon the ethics involved in publishing some types of research data.

> The most interesting thing I think I learned from this was the aspect of ethics involved in releasing research papers. This study involve[d] individuals expressing alleles that resulted in a particular phenotype that resulted from a consanguineous family history. Releasing photos to illustrate these phenotypes (exposing the patient's identity) and writing about the family history could present moral and ethical issues that I think a lot of people don't consider when doing research, although the information conveyed in releasing the photos and family history is beneficial for the reader in understanding the study.
>
> *-male student, senior class*

Collectively, the students therefore reflected on the mechanics of paper publishing and the scientific communication process, the importance of peer review for ensuring high-quality research, and the multiple competing interests involved in both publishing and interpreting research studies.

## Discussion

Preprints are a relatively new entrant into the process of scientific publishing, yet their popularity has soared and has been further catalyzed by the COVID-19 pandemic [14]. Several excellent methods are widely used for teaching students the structure of scientific articles and ways to effectively read articles to maximize understanding [53]. We wanted to add the use of preprint articles into this pedagogical toolbox with the hypothesis that it would help students to be more careful readers of scientific literature and to develop a more sophisticated understanding of the scientific publishing process.

### Preprint articles and the cognitive apprenticeship model of training

To help students with one of the more challenging aspects of reviewing scientific articles, critiquing research data, we used a scaffolded learning approach, employing pairs of preprint and final published articles as pedagogical tools (Fig 4). We first engaged students in a process of mock peer review in which we asked them to critique the results section of the preprint article (part 1 of the assignment). Consistent with the cognitive apprenticeship model, they then identified changes between the preprint and final versions, which were presumably the result of peer review (part 2 of the assignment). This active learning process modeled professional practice for students by revealing the hidden thinking of the peer reviewers. Even though students made a significant number of critiques (average of 2.04) and found changes incorporated

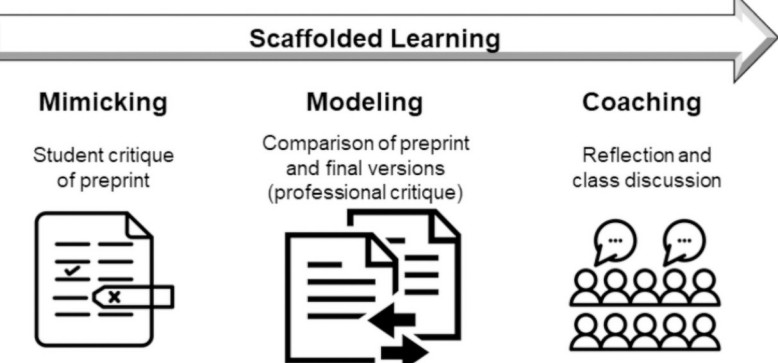

**Fig 4. Scaffolded learning using preprint articles.**

during the peer review process (average of 3.38 changes), few of these suggested and actual modifications addressed the same content or weakness of the article (average of 0.08). By having students compare their critique to defects in the paper identified by professional scientists, students revealed for themselves the differences in how professionals approach the peer review task, consistent with a cognitive apprenticeship approach [29,54,55]. The final stage of cognitive apprenticeship, coaching, was then provided when students engaged in a guided reflection comparing their work to the professionals through both a reflective worksheet (part 3 of the assignment) and through class discussions, as discussed below.

## The benefits of class discussion on this activity

This exercise involved three distinct student activities that were performed sequentially. Following submission of each part of the assignment, the class discussed their results and their impressions, and class discussion is central to this activity for several reasons. First, class discussion is central to ensuring that students properly understand the mechanisms of the peer review process so that misconceptions do not persist. For example, from one student's assignment response, it is unclear whether they understand that there is not a universal time schedule for the peer review process:

> Given that the publication of this paper was completed in under a year, I believe the process of which the paper was published should be maintained if this process can produce accuracy uniformly throughout. What I mean here is that the process for this paper worked, but other studies may not be able to confirm to this time schedule of under a year. This could be troublesome for studies that, again, cannot fill this time crunch and thereby produce results that are not accurate representations of the experiments ran.

*-male student, senior class*

The second reason that class discussion is so valuable is that student responses varied so dramatically during this activity. The range of opinions about whether preprints were a positive development in the publishing process was quite broad, even among students in one class who all read the same paper. Most students had a moderately positive view of preprints, but others were either very strongly in favor or opposed. Most commonly, students agreed that there was a balance between the need to disseminate research results and the importance of ensuring that the work is of the high quality, but they differed in the relative importance that

they assigned to these values, and this can make for a valuable classroom debate. Finally, class discussion is valuable given the wide array of different things that students inferred about the processes of scientific research, data presentation, and the process of scientific publishing and communication (Table 2). Many students had sophisticated insights, including the valuable technique of skipping between sections of a paper when reading, the importance of writing for the reader, that scientists use multiple methods or experimental techniques to bolster a claim, the fact that the meaning of data is not self-evident but must always be interpreted, and that judgment is required in this process. Taken together, these ideas make for a very rich discussion and help students to better understand and appreciate a broad range of concepts related to data presentation and scientific communication.

## The effect of paper selection on this activity

This exercise was conducted at three different universities involving a total of 10 different research articles. Several aspects of papers varied and may have influenced the students' feelings about the use of preprint articles. These included the age of the paper, the number and magnitude of changes that occurred between the preprint and final versions of the paper, and whether the topic involved highly specialized basic research or involved applied research with broad interest, such as work on SARS-CoV2 and COVID-19. Although the total number of students was small, a trend was observed that students who read papers that were older, that had fewer changes, or that involved highly specialized research tended to have a more positive view of preprint articles while those who read papers that were newer, had larger or more numerous changes, or that had the potential for a larger impact were more likely to express concerns about the use of preprints. In some cases, they recognized the importance of preprints, but still expressed concerns about the potential dissemination of data that was "incorrect", perhaps because the results lacked proper controls or required additional experiments. For others, there was concern that the claims made by the paper might be unfounded or might extend beyond what is fair to conclude based on the data. Selection of the paper assigned to students and the degree of change between the preprint and final versions can therefore dramatically influence their first exposure to preprint articles and the peer review process. In one implementation, three students engaging in an independent research course each used a different pair of preprint and final articles. These three articles differed in the number and magnitude of changes between the preprint and final versions; therefore, this led to a robust conversation following each portion of the assignment, particularly after part 2 when students enumerated the changes between the preprint and final versions. A student's initial conclusions based on their assigned paper can therefore be challenged when they hear the results from a different paper which may have gone through a more or less extensive revision process. Assigning more than one paper per class or group can therefore give students a broader and more nuanced understanding of preprints, peer review, and scientific publishing.

## Educating students about the process of science and public communication of science

Science must increasingly cope with the spread of misleading and biased information in what some call a "post-truth" society [56]. Misunderstanding of the process of science and the appropriate role of uncertainty in the scientific process can exacerbate this problem [57]. Key factors in Americans' level of trust in science are their own knowledge level and the sources from which they obtain information, although partisan affiliation also has a strong effect on these results [58]. It is therefore imperative that students have a clear understanding of the scientific research process both so that they can properly interpret scientific information and so

that they can act as conduits of scientific information and understanding to their friends, families, and communities.

In this activity, students learned about diverse aspects of the scientific publishing process, including the nature of peer review and its importance for strengthening the quality of the final published paper (Fig 3). Students also gained a sophisticated understanding of the role of preprint articles and recognized that they could be important drivers of the public conversation around science and medicine [59,60]. They were further able to pick up on the value of preprints in bringing attention both to the preprint itself and to the peer-reviewed article when published [61]. Students also recognized that scientists will be more sophisticated in their understanding of preprints and that these articles need to be taken in context, while media and the general public may have a harder time making this distinction [62].

Impressively, students echoed several of the concerns voiced by scientific researchers about preprint articles including the accelerated pace of peer review during the current pandemic which can imperil quality control [14,63]. Indeed, much like professional scientists, many students pushed back against the demand for faster publication, stressing the need to maintain quality and research integrity [64]. They recognized the increased likelihood that topical research will impact clinical care, and made clear distinctions as to the qualities that should promote publicity for preprint articles, singling out the quality of the research and the public health relevance. Finally, several students made sophisticated arguments about the potential of preprint articles to promote misinformation, an argument that has been echoed by researchers and that is crucial given that 17–30% of SARS-CoV2/COVID-19 papers published in 2020 were preprints [14,65–67]. This exercise therefore had the unexpected and added benefit of helping students to understand the importance of the scientific publication process in conveying accurate information to the public [68] and in preventing and challenging public misconceptions [69], both of which are crucial to the continued development of a scientifically literate society.

## Supporting information

**S1 File. Student self-guided analysis of the introduction, results, and discussion sections of their preprint article.**
(PDF)

**S2 File. Student worksheet for comparison of the preprint and final published version of the article.**
(PDF)

**S3 File. Student worksheet for reflection on the learning experience.**
(PDF)

**S4 File. Example of completed student worksheet (S1 File) for self-guided analysis of the introduction, results, and discussion sections of their preprint article.**
(PDF)

**S5 File. Example of completed student worksheet (S2 File) for comparison of the preprint and final published version of the article.**
(PDF)

## Acknowledgments

We thank the students involved in the implementation and assessment of these materials and our colleagues for valuable discussions about the role and use of preprint articles.

## Author Contributions

**Conceptualization:** Lisa Z. Scheifele, Nikolaos Tsotakos, Michael J. Wolyniak.

**Formal analysis:** Lisa Z. Scheifele, Nikolaos Tsotakos, Michael J. Wolyniak.

**Investigation:** Lisa Z. Scheifele, Nikolaos Tsotakos, Michael J. Wolyniak.

**Methodology:** Lisa Z. Scheifele.

**Project administration:** Lisa Z. Scheifele.

**Writing – original draft:** Lisa Z. Scheifele.

**Writing – review & editing:** Lisa Z. Scheifele, Nikolaos Tsotakos, Michael J. Wolyniak.

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
