## [Decision Letter · Decision Letter 0]

23 Jul 2021

PONE-D-21-18261

PREPRINT ARTICLES AS A TOOL FOR TEACHING DATA ANALYSIS AND SCIENTIFIC COMMUNICATION

PLOS ONE

Dear Dr. Scheifele,

Thank you for submitting your manuscript to PLOS ONE. After careful consideration, we feel that it has merit but does not fully meet PLOS ONE’s publication criteria as it currently stands. Therefore, we invite you to submit a revised version of the manuscript that addresses the points raised during the review process.

We look forward to receiving your revised manuscript.

Kind regards,

Or Kan Soh

Academic Editor

PLOS ONE

Journal Requirements:

Additional Editor Comments:

Dear Author(s)

I am pleased to inform you that the reviewers have come to the conclusions for minor corrections to your manuscript. Please refer to their comments for further action.

Thank you.

Reviewers' comments:

Reviewer's Responses to Questions

**Comments to the Author**

1. Is the manuscript technically sound, and do the data support the conclusions?

Reviewer #1: Yes

Reviewer #2: Yes

Reviewer #3: Yes

2. Has the statistical analysis been performed appropriately and rigorously? 

Reviewer #1: N/A

Reviewer #2: N/A

Reviewer #3: N/A

3. Have the authors made all data underlying the findings in their manuscript fully available?

Reviewer #1: Yes

Reviewer #2: Yes

Reviewer #3: Yes

4. Is the manuscript presented in an intelligible fashion and written in standard English?

Reviewer #1: Yes

Reviewer #2: Yes

Reviewer #3: Yes

5. Review Comments to the Author

Reviewer #1: Preprint articles as a tool for teaching data analysis and scientific communication

In the present article, the authors developed a new method in which students use a cognitive apprenticeship model to uncover how experts analyzed a paper and compare the professional’s cognitive approach to their own. In my opinion, the study is interesting and innovative, including was well delineated.

Reviewer #2: Dear authors, thank you for your collaboration

A very interesting work on teaching strategy.

I would consider making it less extensive in order that readers can locate themselves in the strategy used and the qualitative form that you have used to evaluate the effectiveness of this strategy in Higher Education Teaching.

Reviewer #3: Congratulations on building this model to develop student’s skills in peer review by engaging in realistic, real-work tasks conducted on 28 students from three institutions during the Fall 2020 semester using the cognitive apprenticeship approach.

It is encouraging to observe the teaching of this approach from the earliest stages of university education, which I believe will go a long way to improve peer review and interpreting research data.

I propose to the authors the following questions and suggestions, which might help to clarify some of these concerns:

- Could you describe the methodological design you used?

- Did any students refuse to participate, or did they not complete the entire assessment?

- Regarding the qualitative results, precisely the verbatims, could you provide information about the participant, at least age and gender?

A minor language revision is necessary to improve the manuscript understandable. Below are some grammatical errors to be corrected:

Introduction

Line 25 science to scientific

Line 36 step to steps

Line 37 peer reviewed to peer-reviewed

Line 62 Central to this process are the 5 to Central to this process is the 5

Line 68 critique by having students review articles themselves to critiques by having students review articles themselves

Line 72 difference between their assessment and the peer reviewer’s. to differences between their assessment and the peer reviewers.

Discussion

Line 477 judgement to judgment

6. PLOS authors have the option to publish the peer review history of their article (what does this mean?). If published, this will include your full peer review and any attached files.

Reviewer #1: **Yes: **Tarek Mohamed Abd El-Aziz

Reviewer #2: No

Reviewer #3: **Yes: **Ian Blanco-Mavillard

---

## [Author Response · Author response to Decision Letter 0]

24 Aug 2021

Dear Dr. Soh:

Thank you for your assistance in seeing our manuscript through the peer review process. We appreciate the careful review by the reviewers, their overall strongly positive assessment of our work, and their view that only minor corrections to our manuscript are needed. We have addressed all the reviewers’ requests and include a point-by-point response below.

1. Reviewer #2: I would consider making it less extensive in order that readers can locate themselves in the strategy used and the qualitative form that you have used to evaluate the effectiveness of this strategy in Higher Education Teaching.

While we agree with reviewer 2’s statement that it is necessary for readers to be able to see how they could use or implement this pedagogical strategy, we believe that shortening the manuscript and making it less extensive would hinder that process rather than enabling it. Because our pedagogical technique can be implemented in different classes and contexts, we felt that it is necessary to include sufficient details about each of the different instances of the technique being used that a diverse range of faculty from different institutional types can best use this teaching strategy. 

2. Reviewer #3: Could you describe the methodological design you used?

We appreciate the reviewers’ request here and have included a better explanation that we believe improves the revised manuscript. Specifically, we described how we designed our study methods correspond to three steps of the cognitive apprenticeship approach (lines 113-118 in the revised manuscript with tracked changes), the learning model on which our pedagogical approach is based.

3. Reviewer #3: Did any students refuse to participate, or did they not complete the entire assessment?

No students refused to participate, but this exercise was one of two options that students could choose from in one of the participating courses; we have clarified in the manuscript that 6 of the 24 enrolled students chose to participate in this study (lines 98-99 in the revised manuscript with tracked changes). Two students did not complete the entire assignment, and we have now clarified that student work was not included unless all three parts were completed (lines 101-102 in the revised manuscript with tracked changes).

4. Reviewer #3: Regarding the qualitative results, precisely the verbatims, could you provide information about the participant, at least age and gender?

We appreciate the reviewer’s interest in characteristics of our study participants. We describe the gender and class year of all our participants in the methods section. To respond more precisely to this request while still protecting our students’ privacy, we have included the students’ gender and class year for each of the significant direct quotes, and this information can now be found on lines 257, 281, 323, 391, 429, 480 of the revised manuscript with tracked changes.

5. Reviewer 3: A minor language revision is necessary to improve the manuscript understandable. Below are some grammatical errors to be corrected:

Introduction

Line 25 science to scientific

This correction has been made.

Line 36 step to steps

We believe that this is grammatically correct as written: “publication on a preprint server is 

increasingly becoming the first step of publishing a research article”.

Line 37 peer reviewed to peer-reviewed

This correction has been made.

Line 62 “Central to this process are the” to “Central to this process is the” 

We believe that this is correct as written; the subject of the sentence is “the principles” and therefore the verb should be plural “are”.

Line 68 critique by having students review articles themselves to critiques by having students review articles themselves

This correction has been made.

Line 72 difference between their assessment and the peer reviewer’s. to differences between their assessment and the peer reviewers.

This correction has been made.

Discussion

Line 477 judgement to judgment

This correction has been made.

Thank you again for your help shepherding our manuscript. We hope that these changes have fully addressed the reviewers’ concerns, but if not, we would be happy to make additional revisions as necessary.

Sincerely,

Lisa Scheifele

Associate Professor and Chair

---

## [Decision Letter · Decision Letter 1]

7 Dec 2021

PREPRINT ARTICLES AS A TOOL FOR TEACHING DATA ANALYSIS AND SCIENTIFIC COMMUNICATION

PONE-D-21-18261R1

Dear Dr. Scheifele,

We’re pleased to inform you that your manuscript has been judged scientifically suitable for publication and will be formally accepted for publication once it meets all outstanding technical requirements.

Kind regards,

Muhammad Shahzad Aslam, Ph.D.,M.Phil., Pharm-D

Academic Editor

PLOS ONE

Additional Editor Comments (optional):

Reviewers' comments:

Reviewer's Responses to Questions

**Comments to the Author**

1. If the authors have adequately addressed your comments raised in a previous round of review and you feel that this manuscript is now acceptable for publication, you may indicate that here to bypass the “Comments to the Author” section, enter your conflict of interest statement in the “Confidential to Editor” section, and submit your "Accept" recommendation.

Reviewer #1: (No Response)

Reviewer #2: All comments have been addressed

Reviewer #3: All comments have been addressed

2. Is the manuscript technically sound, and do the data support the conclusions?

Reviewer #1: Yes

Reviewer #2: Yes

Reviewer #3: Yes

3. Has the statistical analysis been performed appropriately and rigorously? 

Reviewer #1: Yes

Reviewer #2: N/A

Reviewer #3: Yes

4. Have the authors made all data underlying the findings in their manuscript fully available?

Reviewer #1: Yes

Reviewer #2: Yes

Reviewer #3: Yes

5. Is the manuscript presented in an intelligible fashion and written in standard English?

Reviewer #1: Yes

Reviewer #2: Yes

Reviewer #3: Yes

6. Review Comments to the Author

Reviewer #1: The authors have been addressed all the comments. I recommend to accept the manuscript in the present version.

Reviewer #2: Dear authors, thank you for your contribution. A very good teaching-learning exercise in sup education

Reviewer #3: The authors correctly respond to the questions of both reviewers. For my part, there are no further methodological concerns or limitations. Congratulations to the authors!

7. PLOS authors have the option to publish the peer review history of their article (what does this mean?). If published, this will include your full peer review and any attached files.

Reviewer #1: No

Reviewer #2: No

Reviewer #3: **Yes: **Ian Blanco-Mavillard

---

## [Editor Report · Acceptance letter]

13 Dec 2021

PONE-D-21-18261R1 

Preprint articles as a tool for teaching data analysis and scientific communication 

Dear Dr. Scheifele:

I'm pleased to inform you that your manuscript has been deemed suitable for publication in PLOS ONE. Congratulations! Your manuscript is now with our production department. 

Kind regards, 

on behalf of

Dr. Muhammad Shahzad Aslam 

Academic Editor

PLOS ONE